# LEARNING CONCISE REPRESENTATIONS FOR REGRESSION BY EVOLVING NETWORKS OF TREES

**William La Cava**,* **Tilak Raj Singh, James P. Taggart, Srinivas Suri, & Jason H. Moore**
Institute for Biomedical Informatics
University of Pennsylvania
`{lacava, moore}@upenn.edu,`
`{tilakraj, jtagg, surisr}@seas.upenn.edu`

## ABSTRACT

We propose and study a method for learning interpretable representations for the task of regression. Features are represented as networks of multi-type expression trees comprised of activation functions common in neural networks in addition to other elementary functions. Differentiable features are trained via gradient descent, and the performance of features in a linear model is used to weight the rate of change among subcomponents of each representation. The search process maintains an archive of representations with accuracy-complexity trade-offs to assist in generalization and interpretation. We compare several stochastic optimization approaches within this framework. We benchmark these variants on 100 open-source regression problems in comparison to state-of-the-art machine learning approaches. Our main finding is that this approach produces the highest average test scores across problems while producing representations that are orders of magnitude smaller than the next best performing method (gradient boosting). We also report a negative result in which attempts to directly optimize the disentanglement of the representation result in more highly correlated features.

## 1 INTRODUCTION

The performance of a machine learning (ML) model depends primarily on the data representation used in training (Bengio et al., 2013), and for this reason the representational capacity of neural networks (NN) is considered a central factor in their success in many applications (Goodfellow et al., 2016). To date, there does not seem to be a consensus on how the architecture should be designed. As problems grow in complexity, the networks proposed to tackle them grow as well, leading to an intractable design space. One design approach is to tune network architectures through network hyperparameters using grid search or randomized search (Bergstra & Bengio, 2012) with cross validation. Often some combination of hyperparameter tuning and manual design by expertise/intuition is done (Goodfellow et al., 2016). Many approaches to network architecture search exist, including weight sharing (Pham et al., 2018) and reinforcement learning (Zoph & Le, 2016). Another potential solution explored in this work (and others) is to use population-based stochastic optimization (SO) methods, also known as metaheuristics (Luke, 2013). In SO, several candidate solutions are evaluated and varied over several iterations, and heuristics are used to select and update the candidate networks until the population produces a desirable architecture. The general approach has been studied at least since the late 80s in various forms (Miller et al., 1989; Yao, 1999; Stanley & Miikkulainen, 2002) for NN design, with several recent applications (Real et al., 2017; Jaderberg et al., 2017; Conti et al., 2017; Real, 2018).

In practice, the adequacy of the architecture is often dependent on conflicting objectives. For example, interpretability may be a central concern, because many researchers in the scientific community rely on ML models not only to provide predictions that match data from various processes, but to provide insight into the nature of the processes themselves. Approaches to interpretability can be roughly grouped into semantic and syntactic approaches. Semantic approaches encompass methods that attempt to elucidate the behavior of a model under various input conditions as a way of explanation (e.g. (Ribeiro et al., 2016)). Syntactic methods instead focus on the development of concise models

---

*Send correspondence to `lacava@upenn.edu`

that offer insight by virtue of their simplicity, in a similar vein to models built from first-principles (e.g. (Tibshirani, 1996; Schmidt & Lipson, 2009)). Akin to the latter group, our goal is to discover the simplest description of a process whose predictions generalize as well as possible.

Good representations should also disentangle the factors of variation (Bengio et al., 2013) in the data, in order to ease model interpretation. Disentanglement implies functional modularity; i.e., sub-components of the network should encapsulate behaviors that model a sub-process of the task. In this sense, stochastic methods such as evolutionary computation (EC) appear well-motivated, as they are premised on the identification and propagation of building blocks of solutions (Holland, 1975). Experiments with EC applied to networks suggest it pressures networks to be modular (Huizinga et al., 2014; Kashtan & Alon, 2005). Although the identification functional building blocks of solutions sounds ideal, we have no way of knowing *a priori* whether a given problem will admit the identification of building blocks of solutions via heuristic search (Oppacher, 2014). Our goal in this paper is thus to empirically assess the performance of several SO approaches in a system designed to produce intelligible representations from NN building blocks for regression.

In Section 2, we introduce a new method for optimizing representations that we call the feature engineering automation tool (FEAT)[1]. The purpose of this method is to optimize an archive of representations that characterize the trade-off between conciseness and accuracy among representations. Algorithmically, two aspects of the method distinguish FEAT from previous work. First, it represents the internal structure of each NN as a set of syntax trees, with the goal of improving the transparency of the resultant representations. Second, it uses weights learned via gradient descent to provide feedback to the variation process at a more granular level. We compare several multi-objective variants of this approach using EC and non-EC methods with different sets of objectives.

We discuss related work in more detail in Section 3. In section 4 and 5, we describe and conduct an experiment that benchmarks FEAT against state-of-the-art ML methods on 100 open-source regression problems. Future work based on this analysis is discussed in Section 6, and additional detailed results are provided in the Appendix.

## 2 METHODS

We are interested in the task of regression, for which the goal is to build a predictive model $\hat{y}(\mathbf{x})$ using $N$ paired examples $\mathcal{T} = \{(\mathbf{x}_i, y_i)\}_{i=1}^N$. The regression model $\hat{y}(\mathbf{x})$ associates the inputs $\mathbf{x} \in \mathbb{R}^d$ with a real-valued output $y \in \mathbb{R}$. The goal of feature engineering / representation learning is to find a new representation of $\mathbf{x}$ via a $m$-dimensional feature mapping $\phi(\mathbf{x}) : \mathbb{R}^d \to \mathbb{R}^m$, such that a model $\hat{y}(\phi(\mathbf{x}))$ outperforms the model $\hat{y}(\mathbf{x})$. We will assume that each predictor in $\mathbf{x}$ is scaled to zero mean, unit-variance.

When applying a NN to a traditional ML task like regression or classification, a fixed NN architecture $\phi(\mathbf{x}, \theta)$, parameterized by $\theta$, is chosen and used to fit a model

$$\hat{y} = \phi(\mathbf{x}, \theta)^T \hat{\beta} \tag{1}$$

In this case $\phi = [\phi_1 \ \dots \ \phi_m]^T$ is a NN representation with $m$ nodes in the final hidden layer and a linear output layer with estimated coefficients $\hat{\beta} = [\hat{\beta}_1 \ \dots \ \hat{\beta}_m]^T$. Typically the problem is then cast as a parameter optimization problem that minimizes a loss function via gradient descent. In order to tune the structure of the representation, we instead wish to solve the joint optimization problem

$$\phi^*(\mathbf{x}, \theta^*) = \arg \min_{\phi \in \mathbb{S}, \theta} \sum_i^N L(y_i, \hat{y}_i, \phi, \theta, \hat{\beta}) \tag{2}$$

where $\hat{\phi}(\mathbf{x}, \hat{\theta})$ is chosen to minimize a cost function $L$, with global optimum $\phi^*(\mathbf{x}, \theta^*)$. ($L$ may depend on $\theta$ and $\beta$ in the case of regularization.) $\mathbb{S}$ is the space of possible representations realizable by the search procedure, and $\phi^*$ is the true structure of the process underlying the data. The assumption of SO approaches such as evolutionary computation (EC) and simulated annealing (SA)

---

[1]http://github.com/lacava/feat

Table 1: Functions and terminals used to develop representations.

| | |
|---|---|
| Continuous functions | $\{+, -, *, /, {}^2, {}^3, \sqrt{}, \sin, \cos, \exp, \log, \text{exponent}, \text{logit}, \tanh, \text{gauss}, \text{relu}\}$ |
| Boolean functions | and, or, not, xor, =, <, <=, >, >= |
| Terminals | $\{\mathbf{x}\}$ |

is that candidate solutions in $\mathbb{S}$ that are similar to each other, i.e. reachable in few mutations, are more likely to have similar costs than candidate solutions that are far apart. In these cases, despite Eqn. 2 being non-convex, $\mathbb{S}$ can be effectively searched by maintaining and updating a population of candidate representations that perform well. Multi-objective SO methods extend Eqn. 2 to minimizing additional cost functions (Deb et al., 2000).

## 2.1 FEAT

FEAT uses a typical $\mu + \lambda$ evolutionary updating scheme, where $\mu = \lambda = P$. The method optimizes a population of potential representations, $\mathbb{N} = \{n_1 \ \dots \ n_P\}$, where $n$ is an "individual" in the population, iterating through these steps:

1. Fit a linear model $\hat{y} = \mathbf{x}^T \hat{\beta}$. Create an initial population $\mathbb{N}$ consisting of this initial representation, $\phi = \mathbf{x}$, along with $P - 1$ randomly generated representations that sample $\mathbf{x}$ proportionally to $\hat{\beta}$.

2. While the stop criterion is not met:
   (a) Select parents $\mathbb{P} \subseteq \mathbb{N}$ using a selection algorithm.
   (b) Apply variation operators to parents to generate $P$ offspring $\mathbb{O}$; $\mathbb{N} = \mathbb{N} \cup \mathbb{O}$
   (c) Reduce $\mathbb{N}$ to $P$ individuals using a survival algorithm.

3. Select and return $n \in \mathbb{N}$ with the lowest error on a hold-out validation set.

Individuals are evaluated using an initial forward pass, after which each representation is used to fit a linear model (Eqn. 1) using ridge regression (Hoerl & Kennard, 1970). The weights of the differentiable features in the representation are then updated using stochastic gradient descent.

### 2.1.1 REPRESENTATION

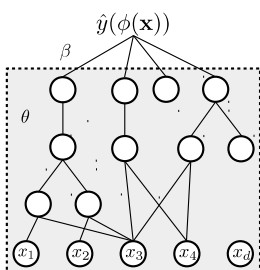

Figure 1: Example model representation in FEAT.

The salient aspect of the proposed method is its use of syntax trees to represent the internal architecture of the network, as shown in Fig. 1. FEAT constructs these trees from elementary boolean- and continuous-valued functions and literals (see Table 1). This scheme is inspired by symbolic regression (SR) (Koza, 1992). In contrast to typical SR, each individual $n$ is a *set* of such trees, the output of which is interpreted as a candidate representation, i.e. $\phi(\mathbf{x}) = [\phi_1 \dots \phi_m]$ for an individual with $m$ trees. The second difference from traditional SR is that the weights of differentiable nodes are encoded in the edges of the graph, rather than as independent nodes. We include instructions typically used as activation functions used in NN, e.g. tanh, sigmoid, logit and relu nodes, elementary arithmetic and boolean operators. Although a fully connected feedforward NN could be represented by this construction, representations in FEAT are biased to be thinly connected. Our hypothesis is that by doing so, we will improve the representation's legibility without sacrificing its capacity for modelling nonlinear relationships.

### 2.1.2 VARIATION

During variation, the representations are perturbed using a set of mutation and crossover methods. FEAT chooses among 6 variation operators that are as follows. *Point mutation* changes a node type to a random one with matching output type and arity. *Insert mutation* replaces a node with a randomly generated depth 1 subtree. *Delete mutation* removes a feature or replaces a sub-program with an

input node, with equal probability. *Insert/Delete dimension* adds or removes a new feature. *Sub-tree crossover* replaces a sub-tree from one parent with the sub-tree of another parent. *Dimension crossover* swaps two features between parents. The exact probabilities of each variation operator will affect the performance of the algorithm, and others have proposed methods for adjusting these probabilities, e.g. Igel & Kreutz (2003). For the purposes of our study, we use each operator with uniform probability.

**Feedback** The use of an ML model to assess the fitness of each representation can be used to provide information about the elements of the representation that should be changed. In particular, we assume that programs in the representation with small coefficients are the best candidates for mutation and crossover. With this in mind, let $n$ be an $m$-dimensional candidate representation with associated coefficients $\beta(n) \in \mathbb{R}^m$. Let $\tilde{\beta}_i(n) = |\beta_i| / \sum_i^m |\beta_i|$. The probability of mutation for tree $i$ in $n$ is denoted $PM_i(n)$, and defined as follows:

$$
\begin{aligned}
s_i(n) &= \exp(1 - \tilde{\beta}_i) / \sum_i^m \exp(1 - \tilde{\beta}_i) \\
PM_i(n) &= f s_i(n) + (1 - f)\frac{1}{m}
\end{aligned}
\tag{3}
$$

The normalized coefficient magnitudes $\tilde{\beta} \in [0, 1]$ are used to define softmax-normalized probabilities, $s$ in Eqn. 3. The smaller the coefficient, the higher the probability of mutation. The parameter $f$ is used to control the amount of feedback used to weight the probabilities; $\frac{1}{m}$ in this case represents uniform probability. Among nodes in tree $m$, mutation occurs with uniform probability. This weighting could be extended for differentiable nodes by weighting the within-tree probabilities by the magnitude of the weights associated with each node. However we expect this would yield diminishing returns.

### 2.1.3 SELECTION AND SURVIVAL

The selection step selects among $P$ parents those representations that will be used to generate offspring. Following variation, the population consists of $2P$ representations of parents and offspring. The survival step is used to reduce the population back to size $P$, at which point the generation is finished. In our initial study, we empirically compared five algorithms for selection and survival: 1) $\epsilon$-lexicase selection (Lex) (La Cava et al., 2016b), 2) non-dominated sorting genetic algorithm (NSGA2) (Deb et al., 2000), 3) a novel hybrid algorithm using Lex for selection and NSGA2 for survival, 4) simulated annealing (Kirkpatrick et al., 1983), and 5) random search. These comparisons are described in Appendix Section A.2. We found that the hybrid algorithm (3) performed the best; it is described below.

Parents are selected using Lex. Lex was proposed for regression problems (La Cava et al., 2016b; 2018a) as an adaption of lexicase selection (Spector, 2012) for continuous domains. Under $\epsilon$-lexicase selection, parents are chosen by filtering the population according to randomized orderings of training samples with the $\epsilon$ threshold defined relative to the sample loss among the selection pool. This filtering strategy scales probability of selection for an individual based on the difficulty of the training cases the individual performs well on. Lex has shown strong performance among SR methods in recent tests, motivating our interest in studying it (Orzechowski et al., 2018). The survival step for Lex just preserves offspring plus the best individual in the population.

Survival is conducted using the survival sub-routine of NSGA2, a popular strategy for multi-objective optimization (Deb et al., 2000). NSGA2 applies preference for survival using Pareto dominance relations. An individual $(n_i)$ is said to *dominate* another $(n_j)$ if, for all objectives, $n_i$ performs at least as well as $n_j$, and for at least one objective, $n_i$ strictly outperforms $n_j$. The Pareto *front* is the set of individuals in $\mathbb{N}$ that are non-dominated in the population and thus represent optimal trade-offs between objectives found during search. Individuals are assigned a Pareto *ranking* that specifies the number of individuals that dominate them, thereby determining their proximity to the front.

The survival step of NSGA2 begins by sorting the population according to their Pareto front ranking and choosing the lowest ranked individuals for survival. To break rank ties, NSGA2 assigns each

individual a crowding distance measure, which quantifies an individual's distance to its two adjacent neighbors in objective space. If a rank level does not completely fit in the survivor pool, individuals of that rank are sorted by highest crowding distance and added in order until $P$ individuals are chosen.

### 2.1.4 OBJECTIVES

We consider three objectives in our study corresponding to three goals: first, to reduce model error; second, to minimize complexity of the representation; and third, to minimize the entanglement of the representation. We test the third objective using two different metrics: the correlation of the transformation matrix $\phi(\mathbf{x})$ and its condition number. These metrics are defined below.

The first objective always corresponds to the mean squared loss function for individual $n$, and the second corresponds to the complexity of the representation. There are many ways to define complexity of an expression; one could simply look at the number of operations in a representation, or look at the behavioral complexity of the representation using a polynomial order (Vladislavleva et al., 2009). The one we use, which is similar to that used by Kommenda et al. (2015), is to assign a complexity weight to each operator (see Table 1), with higher weights assigned to operators considered more complex. If the weight of operator $o$ is $c_o$, then the complexity of an expression tree beginning at node $o$ is defined recursively as

$$C(o) = c_o \sum_{a=1}^{k} C(a) \tag{4}$$

where $o$ has $k$ arguments, and $C(a)$ is the complexity of argument $a$. The complexity of a representation is then defined as the sum of the complexities of its output nodes. The goal of defining complexity in such a way is to discourage deep sub-expressions within complex nodes, which are often hard to interpret. It's important to note that the choice of operator weights is bound to be subjective, since we lack an objective notion of interpretability. For this reason, although we use Eqn. 4 to drive search, our experimental comparisons with other algorithms rely on the node counts of the final models for benchmarking interpretability of different methods.

*Disentanglement* is a term used to describe the notion of a representation's ability to separate factors of variation in the underlying process (Bengio et al., 2013). Although a thorough review is beyond the scope of this section, there is a growing body of literature addressing disentanglement, primarily with unsupervised learning and/or image analysis (Montavon & Müller, 2012; Whitney, 2016; Higgins et al., 2017; Gonzalez-Garcia et al., 2018; Hadad et al., 2018; Kumar et al., 2018). There are various ways to quantify disentanglement. For instance, Brahma et al. (2016) proposed measuring disentanglement as the difference between geodesic and Euclidean distances among points on a manifold (i.e. training instances). If the latent structure is known, the information-theoretic metrics proposed by Eastwood & Williams (2018) may be used. In the case of regression, a disentangled representation ideally contains a minimal set of features, each corresponding to a separate latent factor of variation, and each orthogonal to each other. In this regard, we attempt to minimize the collinearity between features in $\phi$ as a way to promote disentanglement. We tested two measurements of collinearity (a.k.a. multicollinearity) in the derived feature space. The first is the average squared Pearson's correlation among features of $\phi$, i.e.,

$$Corr(\phi) = \frac{1}{N(N-1)} \sum_{\phi_i, \phi_j \in \phi, i \neq j} \left( \frac{\mathrm{cov}(\phi_i, \phi_j)}{\sigma(\phi_i)\sigma(\phi_j)} \right)^2 \tag{5}$$

Eqn. 5 is relatively inexpensive to compute but only captures bivariate correlations in $\phi$. As a result we also test the condition number (CN). Consider the $N \times m$ representation matrix $\Phi$. The CN of $\Phi$ is defined as

$$CN(\phi) = \frac{\mu_{\max}(\Phi)}{\mu_{\min}(\Phi)} \tag{6}$$

where $\mu_{\max}$ and $\mu_{\min}$ are the largest and smallest singular values of $\Phi$. Unlike $Corr$, $CN$ can capture higher-order dependencies in the representation. $CN$ is also related directly to the sensitivity of $\Phi$ to perturbations in the training data (Belsley, 1991; Cline et al., 1979), and thus captures a notion of network invariance explored in previous work by Goodfellow et al. (2009). We consider another common measure of multicollinearity, the variance inflation factor (O'brien, 2007), to be too expensive for our purposes.

## 3 RELATED WORK

The idea to evolve NN architectures is well established in literature, and is known as neuroevolution. Popular methods of neuroevolution include neuroevolution of augmenting topologies (NEAT(Stanley & Miikkulainen, 2002) and Hyper-NEAT(Stanley et al., 2009)), and compositional pattern producing networks (Stanley, 2007) . The aforementioned approaches eschew the parameter learning step common in other NN paradigms, although others have developed integrations (Fernando et al., 2016). In addition, they have been developed predominantly for other task domains such as robotics and control (Gomez et al., 2006), image classification (Real et al., 2017; Real, 2018), and reinforcement learning (Igel, 2003; Conti et al., 2017). Reviews of these methods are available (Floreano et al., 2008; Stanley et al., 2019).

Most neuroevolution strategies do not have interpretability as a core focus, and thus do not attempt to use multi-objective methods to update the networks. An exception is the work of Wiegand et al. (2004), in which a template NN was optimized using a multi-objective EC method with size as an objective. In this case, the goal was to reduce computational complexity in face detection.

Neuroevolution is a part of a broader research field of neural architecture search (NAS) (Zoph & Le, 2016; Le & Zoph, 2017; Liu et al., 2017). NAS methods vary in approach, including for example parameter sharing (Pham et al., 2018), sequential model-based optimization (Liu et al., 2017), reinforcement learning (Zoph & Le, 2016), and greedy heuristic strategies (Cortes et al., 2016).

FEAT is also related to SR approaches to feature engineering (Krawiec, 2002; Arnaldo et al., 2014; La Cava & Moore, 2017; La Cava et al., 2018b; Muñoz et al., 2018) that use EC to search for possible representations and couple with an ML model to handle the parametrization of the representations. SR methods have been successful in developing intelligible models of physical systems (Schmidt & Lipson, 2009; La Cava et al., 2016a). FEAT differs from these methods in the following ways. A key challenge in SR is understanding functional modularity within representations/programs that can be exploited for search. FEAT is designed with the insight that ML weights can be leveraged during variation to promote functional building blocks, an exploit not used in previous methods. Second, FEAT uses multiple type representations, and thus can learn continuous and rule-based features within a single representation, unlike previous methods. This is made possible using a stack-based encoding with strongly-typed operators. Finally, FEAT incorporates two elements of NN learning to improve its representational capacity: activation functions commonly used in NN and edge-based encoding of weights. Traditionally, SR operates with standard mathematical operators, and treats constants as leaves in the expression trees rather than edge weights. An exception is MRGP (Arnaldo et al., 2014), which encodes weights at each node but updates them via Lasso instead of using gradient descent with backpropagation. SR methods have also been paired with various parameter learning strategies, including those based on backpropagation (Topchy & Punch, 2001; Kommenda et al., 2013; Izzo et al., 2017). It should be noted that non-stochastic methods for SR exist, such as mixed integer non-linear programming, which has been demonstrated for small search spaces (Austel et al., 2017).

## 4 EXPERIMENT

Our goals with the experiment are to 1) robustly compare FEAT to state-of-the-art regression methods, including hyperparameter optimization of feedforward NNs; 2) characterize the complexity of the models; and 3) assess whether disentanglement objectives lead to less correlated representations. For the regression datasets, we use 100 real-world and simulated datasets available from OpenML (Vanschoren et al., 2014). The datasets are characterized in terms of number of features and sample sizes in Figure 2. We use the standardized versions of the datasets available in the Penn Machine Learning Benchmark repository (Olson et al., 2017). We compare the FEAT configurations to multi-layer perceptron (MLP), random forest (RF) regression, kernel ridge (KernelRidge) regression, and elastic net (ElasticNet) regression, using implementations from scikit-learn (Pedregosa et al., 2011). In addition, we compare to XGBoost (XGB), a gradient boosting method that has performed well in recent competitions (Chen & Guestrin, 2016). Code to reproduce these experiments is available online.[2]

---

[2]https://github.com/lacava/iclr_2019

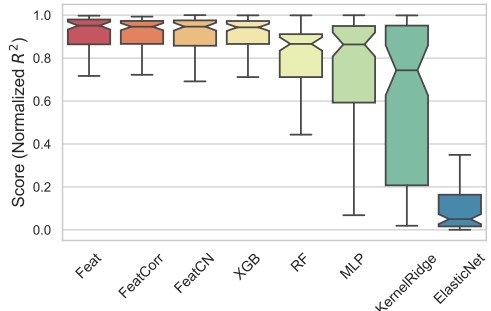

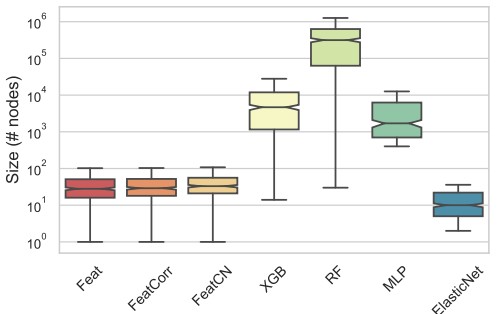

Figure 3: Mean 10-fold CV $R^2$ performance for various SO methods in comparison to other ML methods, across the benchmark problems.

Figure 4: Size comparisons of the final models in terms of number of nodes in the solutions.

Each method's hyperparameters are tuned according to Table 2 in Appendix A.1. For FEAT, we limit optimization to 200 iterations or 60 minutes, whichever comes first. We also stop sooner if the median validation fitness stops improving. For each method, we use grid search to tune the hyperparameters with 10-fold cross validation (CV). We use the $R^2$ CV score for assessing performance. In our results we report the CV scores for each method using its best hyperparameters. The algorithms are ranked on each dataset using their median CV score over 5 randomized shuffles of the dataset. For comparing complexity, we count the number of nodes in the final model produced by each method for each trial on each dataset. To quantify the "entanglement" of the feature spaces, we report Eqn. 5 in the raw data and in the final hidden layer of FEAT and MLP models. We also test two additional versions of Feat, denoted FeatCorr and FeatCN, that include a third objective corresponding to Eqn. 5 and 6, respectively.

Finally, we examine the FEAT results in detail for one of the benchmark datasets. For this dataset we plot the final population of models, illustrate model selection and compare the resultant features to results from linear and ensemble tree-based results. This gives practical insight into the method and provides a sense of the intelligibility of an example representation.

## 5   RESULTS

The score statistics for each method are shown in Fig. 3 across all datasets. Full statistical comparisons are reported in Appendix A.3. Over all, FEAT and XGBoost produce the best predictive performance across datasets without significant differences between the two ($p$=1.0). FEAT significantly outperforms MLP, RF, KernelRidge and ElasticNet ($p \leq$1.18e-4), as does XGBoost ($p \leq$1.6e-3).

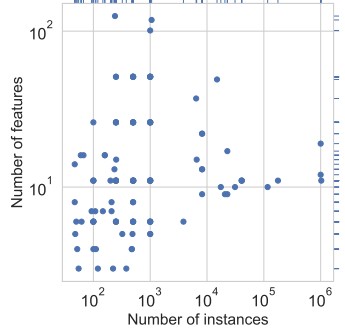

Figure 2: Properties of the regression benchmarks.

As measured by the number of nodes in the final solutions, the models produced by FEAT are significantly less complex than XGBoost, RF, and MLP, as shown in Fig. 4 ($p$<1e-16). FEAT's final models tend to be within 1 order of magnitude of the linear models (ElasticNet), and 2-4 orders of magnitude smaller than the other non-linear methods.

A comparison of wall-clock time is given in Fig. 7 in the appendix. FEAT and MLP take approximately the same time to run, followed by XGBoost, RF, KernelRidge, and ElasticNet, in that order.

Fig. 5 shows the average pairwise correlations of the representations produced by Feat variants and MLP in comparison to the correlation structure of the original data. In general, MLP and FEAT tend to produce correlated feature spaces, and Feat's representations tend to contain more bivariate correlations than MLP. Furthermore, the results suggest that explicitly minimizing collinearity

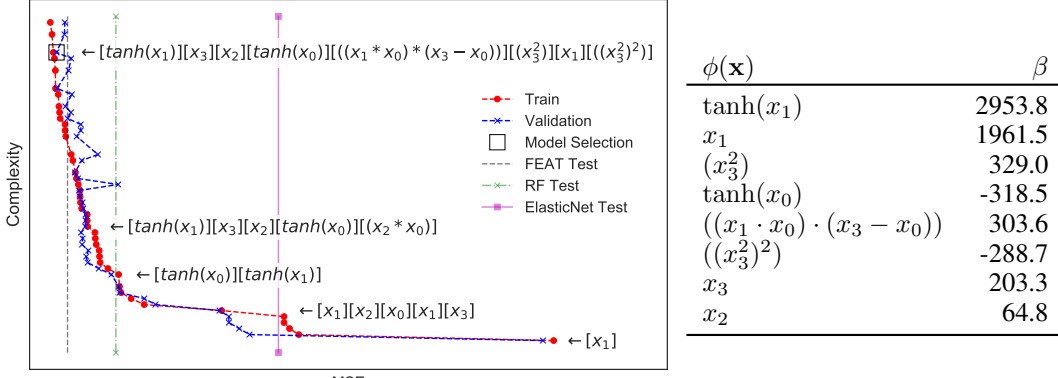

| $\phi(\mathbf{x})$ | $\beta$ |
|---|---:|
| $\tanh(x_1)$ | 2953.8 |
| $x_1$ | 1961.5 |
| $(x_3^2)$ | 329.0 |
| $\tanh(x_0)$ | -318.5 |
| $((x_1 \cdot x_0) \cdot (x_3 - x_0))$ | 303.6 |
| $((x_3^2)^2)$ | -288.7 |
| $x_3$ | 203.3 |
| $x_2$ | 64.8 |

Figure 6: (Left) Representation archive for the visualizing galaxies dataset. (Right) Selected model and its weights. Internal weights omitted.

(FeatCorr and FeatCN) tends to produce representations that exhibit equivalent or higher levels of correlation. This result conflicts with our hypothesis, and is discussed more in Section 6.

**Illustrative Example** We show an illustrative example of the final archive and model selection process from applying FEAT to a galaxy visualization dataset (Cleveland, 1993) in Figure 6. The red and blue points correspond to training and validation scores for each archived representation with a square denoting the final model selection. Five of the representations are printed in plain text, with each feature separated by brackets. The vertical lines in the left figure denote the test scores for FEAT, RF and ElasticNet. It is interesting to note that ElasticNet performance roughly matches the performance of a linear representation, and the RF test performance corresponds to the representation $[\tanh(x_0)][\tanh(x_1)]$ that is suggestive of axis-aligned splits for $x_0$ and $x_1$. The selected model is shown on the right, with the features sorted according to the magnitudes of $\beta$ in the linear model. The final representation combines tanh, polynomial, linear and

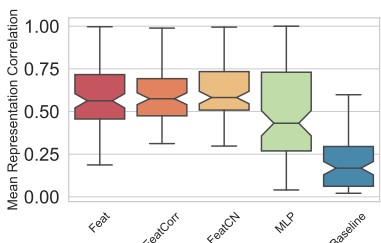

Figure 5: Mean pairwise correlations for representations produced by different methods. Baseline refers to the correlations in the original data.

interacting features. This representation is a clear extension of simpler ones in the archive, and the archive thereby serves to characterize the improvement in predictive accuracy brought about by increasing complexity. Although a mechanistic interpretation requires domain expertise, the final representation is certainly concise and amenable to interpretation.

## 6 DISCUSSION AND CONCLUSION

This paper proposes a feature engineering archive tool that optimizes neural network architectures by representing them as syntax trees. FEAT uses model weights as feedback to guide network variation in an EC optimization algorithm. We conduct a thorough analysis of this method applied to the task of regression in comparison to state-of-the-art methods. The results suggest that FEAT achieves state-of-the-art performance on regression tasks while producing representations that are significantly less complex than those resulting from similarly performing methods. This improvement comes at an additional computational cost, limited in this study to 60 minutes per training instance. We expect this limitation to be reasonable for many applications where intelligibility is the prime motivation.

Future work should consider the issue of representation disentanglement in more depth. Our attempts to include additional search objectives that explicitly minimize multicollinearity were not successful. Although more analysis is needed to confirm this, we suspect that the model selection procedure (Section 2.1, step 3) permits highly collinear representations to be chosen. This is because

multicollinearity primarily affects the standard errors of $\hat{\beta}$ (Belsley, 1991), and is not necessarily detrimental to validation error. Therefore it could be incorrect to expect the model selection procedure to effectively choose more disentangled representations. Besides improving the model selection procedure, it may be fruitful to pressure disentanglement at other stages of the search process. For example, the variation process could prune highly correlated features, or the disentanglement metric could be combined with error into a single loss function with a tunable parameter. We hope to pursue these ideas in future studies.

## 7 ACKNOWLEDGMENTS

This work was supported by NIH grants AI116794 and LM012601.

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

# A  APPENDIX

## A.1  ADDITIONAL EXPERIMENT INFORMATION

Table 2 details the hyperparameters for each method used in the experimental results described in Sections 4 and 5.

Runs are conducted in a heterogenous computing environment, with one core assigned to each CV training per dataset. As such, wall-clock times are a flawed measure of computational complexity. With this caveat in mind, we report the wall-clock run times for each method in Fig. 7. The Feat variants are terminated at 200 generations or 60 minutes, which explains their uniformity. Note that methods are unable to take advantage of parallelization in this experiment.

Table 2: Comparison methods and their hyperparameters. Tuned values denoted with brackets.

| Method | Setting | Value |
|---|---|---|
| FEAT | Population size | 500 |
| | Termination criterion | 200 generations, 60 minutes, or 50 iterations of stalled median validation loss |
| | Max depth | 10 |
| | Max dimensionality | 50 |
| | Objectives | $\{(MSE,C), (MSE,C,Corr),(MSE,C,CN)\}$ |
| | Feedback ($f$) | $\{ 0.25, 0.5, 0.75 \}$ |
| | Crossover/mutation ratio | $\{ 0.25, 0.5, 0.75 \}$ |
| | Batch size | 1000 |
| | Learning rate (initial) | 0.1 |
| | SGD iterations / individual / generation | 10 |
| MLP | Optimizer | {LBFGS, Adam (Kingma & Ba, 2014)} |
| | Hidden Layers | {1,3,6} |
| | Neurons | {(100,), (100,50,10), (100,50,20,10,10,8)} |
| | Learning rate | (initial) {1e-4, 1e-3, 1e-2} |
| | Activation | {logistic, tanh, relu} |
| | Regularization | $L_2, \alpha = $ {1e-5, 1e-4, 1e-3} |
| | Max Iterations | 10000 |
| | Early Stopping | True |
| XGBoost | Number of estimators | {10, 100, 200, 500, 1000} |
| | Max depth | {3, 4, 5, 6, 7} |
| | Min split loss ($\gamma$) | {1e-3,1e-2,0.1,1,10,1e2,1e3} |
| | Learning rate | $\{0, 0.01, \dots, 1.0 \}$ |
| Random Forest | Number of estimators | {10, 100, 1000} |
| | Min weight fraction leaf | $\{ 0.0, 0.25, 0.5 \}$ |
| Kernel Ridge | Kernel | Radial basis function |
| | Regularization ($\alpha$) | $\{ 1e-3, 1e-2, 0.1, 1 \}$ |
| | Kernel width ($\gamma$) | $\{ 1e-2, 0.1, 1, 10, 100 \}$ |
| ElasticNet | $l_1$-$l_2$ ratio | $\{ 0, 0.01, \dots, 1.0 \}$ |
| | selection | { cyclic, random } |

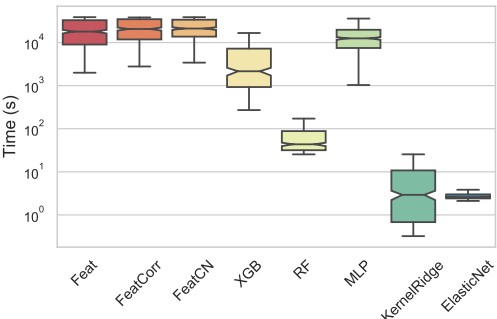

Figure 7: Wall-clock runtime for each method in seconds.

## A.2  COMPARISON OF STOCHASTIC OPTIMIZATION APPROACHES

Our initial analysis sought to determine how different SO approaches performed within this framework. We tested five methods: 1) NSGA2, 2) Lex, 3) LexNSGA2, 4) Simulated annealing, and 5) random search. The simulated annealing and random search approaches are described below.

**Simulated annealing**  Simulated annealing (SimAnn) is a non-evolutionary technique that instead models the optimization process on the metallurgical process of annealing. In our implementation, offspring compete with their parents; in the case of multiple parents, offspring compete with the

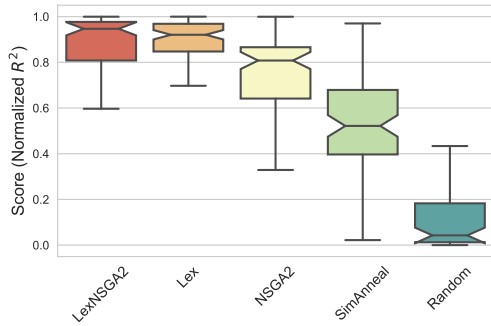 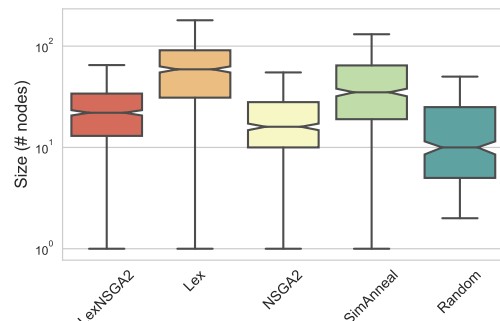

Figure 8: Mean 10-fold CV $R^2$ performance for various SO methods in comparison to other ML methods, across the benchmark problems.

Figure 9: Size comparisons of the final models in terms of number of parameters.

program with which they share more nodes. The probability of an offspring replacing its parent in the population is given by the equation

$$P_{sel}(n_o|n_p, t) = \exp\left(\frac{F(n_p) - F(n_o)}{t}\right) \tag{7}$$

The probability of offspring replacing its parent is a function of its fitness, $F$, in our case the mean squared loss of the candidate model. In Eqn. 7, $t$ is a scheduling parameter that controls the rate of "cooling", i.e. the rate at which steps in the search space that are worse are tolerated by the update rule. In accordance with (Kirkpatrick et al., 1983), we use an exponential schedule for $t$, defined as $t_g = (0.9)^g t_0$ , where $g$ is the current generation and $t0$ is the starting temperature. $t0$ is set to 10 in our experiments.

**Random search** We compare the selection and survival methods to random search, in which no assumptions are made about the structure of the search space. To conduct random search, we randomly sample $\mathbb{S}$ using the initialization procedure. Since FEAT begins with a linear model of the process, random search will produce a representation at least as good as this initial model on the internal validation set.

**A note on archiving** When FEAT is used without a complexity-aware survival method (i.e., with Lex, SimAnn, Random), a separate population is maintained that acts as an archive. The archive maintains a Pareto front according to minimum loss and complexity (Eqn 4). At the end of optimization, the archive is tested on a small hold-out validation set. The individual with the lowest validation loss is the final selected model. Maintaining this archive helps protect against overfitting resulting from overly complex / high capacity representations, and also can be interpreted directly to help understand the process being modelled.

We benchmarked these approaches in a separate experiment on 88 datasets from PMLB (Olson et al., 2017). The results are shown in Figures 8-11. Considering Figures 8 and 9, we see that LexNSGA2 achieves the best average $R^2$ value while producing small solutions in comparison to Lex. NSGA2, SimAnneal, and Random search all produce less accurate models. The runtime comparisons of the methods in Figure 10 show that they are mostly within an order of magnitude, with NSGA2 being the fastest (due to its maintenance of small representations) and Random search being the slowest, suggesting that it maintains large representations during search. The computational behavior of Random search suggests the variation operators tend to increase the average size of solutions over many iterations.

### A.3 STATISTICAL COMPARISONS

We perform pairwise comparisons of methods according to the procedure recommended by Demšar (2006) for comparing multiple estimators. In Table 3, the CV $R^2$ rankings are compared. In

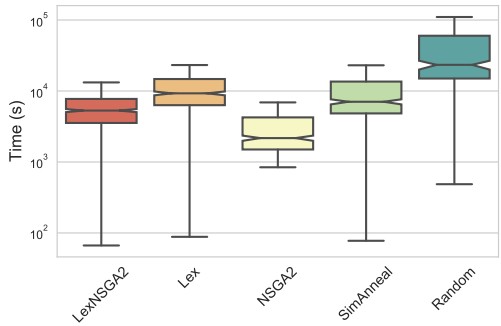

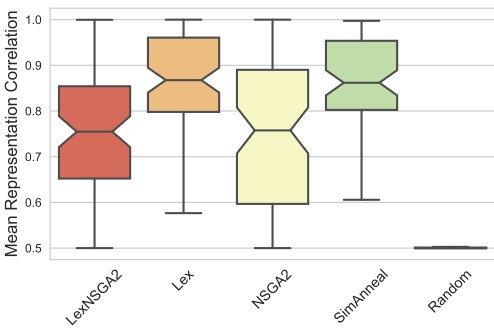

Figure 10: Wall-clock runtime for each method in seconds.

Figure 11: Mean correlation between engineered features for different SO methods compared to the correlations in the original data (ElasticNet).

Table 4, the best model size rankings are compared. Note that KernelRidge is omitted from the size comparisons since we don't have a comparable way of measuring the model size.

Table 3: Bonferroni-adjusted $p$-values using a Wilcoxon signed rank test of $R^2$ scores for the methods across all benchmarks. *: $p<0.05$.

|  | ElasticNet | Feat | FeatCN | FeatCorr | KernelRidge | MLP | RF |
|---|---|---|---|---|---|---|---|
| Feat | 4.70e-14* | | | | | | |
| FeatCN | 1.38e-12* | 5.34e-01 | | | | | |
| FeatCorr | 4.25e-13* | 1.00e+00 | 1.00e+00 | | | | |
| KernelRidge | 1.16e-09* | 1.18e-04* | 4.37e-03* | 1.14e-03* | | | |
| MLP | 5.24e-09* | 3.80e-04* | 2.08e-02* | 1.28e-03* | 1.00e+00 | | |
| RF | 1.08e-09* | 2.09e-07* | 2.19e-05* | 1.30e-06* | 1.00e+00 | 1.00e+00 | |
| XGB | 1.47e-13* | 1.00e+00 | 1.00e+00 | 1.00e+00 | 3.41e-04* | 1.60e-03* | 8.49e-13* |

Table 4: Bonferroni-adjusted $p$-values using a Wilcoxon signed rank test of sizes for the methods across all benchmarks. All results are significant. *: $p<0.05$.

|  | ElasticNet | Feat | FeatCN | FeatCorr | MLP | RF |
|---|---|---|---|---|---|---|
| Feat | 8.25e-12* | | | | | |
| FeatCN | 2.47e-16* | 2.12e-08* | | | | |
| FeatCorr | 1.37e-12* | 1.00e+00 | 1.58e-07* | | | |
| MLP | 6.24e-18* | 4.26e-17* | 3.09e-17* | 3.98e-17* | | |
| RF | 9.28e-20* | 2.05e-17* | 5.61e-18* | 2.37e-17* | 3.54e-17* | |
| XGB | 9.14e-18* | 4.05e-17* | 2.46e-17* | 3.71e-17* | 1.00e+00 | 3.94e-18* |

