# OpenReview forum: "Learning concise representations for regression by evolving networks of trees"
_ICLR.cc/2019/Conference_

### Official Review · AnonReviewer2 · 2018-11-01
**Interesting method with very promising results.**

**Rating:** 8
**Confidence:** 4

**Review:**

The paper proposes a method for learning regression models through evolutionary
algorithms that promise to be more interpretable than other models while
achieving similar or higher performance. The authors evaluate their approach on
99 datasets from OpenML, demonstrating very promising performance.

The authors take a very interesting approach to modeling regression problems by
constructing complex algebraic expressions from simple building blocks with
genetic programming. In particular, they aim to keep the constructed expression
as small as possible to be able to interpret it easier. The evaluation is
thorough and convincing, demonstrating very good results.

The presented results show that the new method beats the performance of existing
methods; however, as only very limited hyperparameter tuning for the other
methods was performed, it is unclear to what extent this will hold true in
general. As the main focus of the paper is on the increased interpretability of
the learned models, this is only a minor flaw though.

The interpretability of the final models is measured in terms of their size.
While this is a reasonable proxy that is easy to measure, the question remains
to what extent the models are really interpretable by humans. This is definitely
something that should be explored in future work, as a small-size model does not
necessarily imply that humans can understand it easily, especially as the
generated algebraic expressions can be complex even for small trees.

The description of the proposed method could be improved; in particular it was
unclear to this reviewer why the features needed to be differentiable and what
the benefit of this was (i.e. why was this the most appropriate way of adjusting
weights).

In summary, the paper should be accepted.

---

> ### Author Response · Authors · 2018-11-15
> **Revision expands hyperparameters, and a note on the choice of constant optimization**
>
> We thank the reviewer for their comments, and address a few minor points below.
>
> 1) "only very limited hyperparameter tuning for the other methods was performed"
>
>   - We have extended the hyperparameter space for XGBoost, the closest competitor, in our revision. Hopefully this addresses the reviewer's concern.
>
> 2) The reviewer correctly points out that size is only a proxy for interpretability in this experiment. We do not have a better way to assess lebility outside of an application with expert analysis. Nevertheless, simpler models are generally (but not always) easier to interpret. Our goal with the illustrative example is to show this, and we state similar caveats as the reviewer has suggested.
>
> 3) Regarding adjustment of weights, weights are only adjusted for features that are composed of differentiable operators because this is a limitation of the chain rule with gradient descent. It is important to note that all of the floating point operators we considered were differentiable; the only non-differentiable nodes were boolean operators, which don't include weights. It would also be possible to use another method to tune the weights such as stochastic hillclimbing, although previous symbolic regression research on this subject tends to favor gradient descent for weight tuning weights [1,2]. Hopefully this addresses the reviewer's question; if not we are happy to clarify further.
>
> [1] Kommenda, M. et. al. (2013, July). Effects of constant optimization by nonlinear least squares minimization in symbolic regression. In Proceedings of the 15th annual conference companion on Genetic and evolutionary computation (pp. 1121-1128). ACM.
> [2] Topchy, A., & Punch, W. F. (2001, July). Faster genetic programming based on local gradient search of numeric leaf values. In Proceedings of the 3rd Annual Conference on Genetic and Evolutionary Computation (pp. 155-162). Morgan Kaufmann Publishers Inc..

---

### Official Review · AnonReviewer3 · 2018-11-02
**This paper lacks technical novelty and experimental result is incomplete.**

**Rating:** 6
**Confidence:** 3

**Review:**

This paper introduces a genetic algorithm that maintains an archive of representations that are iteratively evolved and selected by comparing validation error. Each representation is constructed as a syntax tree consists of elements that are common in neural network architectures. The experimental results showed that their algorithm is competitive to the state-of-the-art while achieving much smaller model size.

Comments:
1. I think this paper lacks technical novelty. I'm going to focus on experimental result in the following two questions.
2. FEAT is a typical genetic algorithm that converges slowly. In the appendix, one can verify that FEAT converges at least 10x slower than XGBoost. Can FEAT achieve lower error than XGBoost when they use the same amount of time?
Can the authors provide a convergence plot of their algorithm (i.e. real time vs test error)?
3. From Figure 3 it seems that the proposed algorithm is competitive to XGBoost, and the model size is much smaller than XGBoost. Have the authors tried to post-processing the model generated by XGBoost? How's the performance compare?

---

> ### Author Response · Authors · 2018-11-15
> **On the novelty of this work, and a discussion of updated experiments**
>
> We thank the reviewer for the critiques, which have led to some improvements to our experiment and hopefully more convincing analysis.
>
> 1. It is hard for us to respond to the reviewer's contention that our work lacks technical novelty without more specific critiques. However, we will restate what is novel here.
>
> First, FEAT represents models in the population as sets of syntax trees/equations. This representation is novel both in neural network literature and genetic algorithm literature. Second, we use the feedback of model weights to guide variation probabilities; to our knowledge this is a new approach. FEAT also uses multiple type representations, meaning it can learn boolean and continuous functions in the same representation, something we believe to be novel as well. Finally, the composition of syntax trees using NN activation functions along with other operations is rarely seen in GA/GP literature, much less the edge-based encoding of weights. Taken as a whole, there are several novel technical aspects of the algorithm.
>
> In addition to the methodological aspects, few if any previous works in neural architecture search / neuroevolution focus on regression with the goal of intelligibility. In this regard we believe our results are novel and important: by establishing a new state-of-the-art, they point to a new area of application for this field of research.
>
> 2. We completely agree with the reviewer's point that FEAT converges more slowly than XGBoost. We should expect a randomized, population-based heuristic search method to be slower than a greedy, single-model heuristic-based method. To address this point, we have added text to the experiments and discussion, and reworked the XGBoost analysis .
>
> Our stated goal is to produce simplest possible models without sacrificing accuracy, and we contend that our method achieves this. Although computation time suffers as a result, we believe it is reasonable to consider a 60 minute cutoff for optimization time on every problem, some of which contain millions of samples.
>
> The reviewer also asks whether FEAT can achieve lower error than XGBoost given the same amount of time. Based on the reviewer's comments we have expanded the hyperparameter space for XGBoost in our revision, from 9 hyperparameter combinations to 1925. This extension results in wallclock runtimes closer to those of FEAT and MLP. Under these conditions, the accuracy comparisons do not change much. We still see no significant differences between FEAT and XGBoost in terms of accuracy.
>
> 3. To address the reviewer's suggestion regarding complexity, we have generated our XGBoost results in this revision with a pruning step after tree construction. We have also optimized the minimum split loss criterion (gamma) that controls the amount of pruning. Under these conditions, we observe very similar size comparisons as before.
>
> We hope the updated manuscript addresses the reviewer's concerns.

---

> > ### Comment · AnonReviewer3 · 2018-11-19
> > **Some Updates after reading authors' comments and other reviews**
> >
> > 1. I should say I'm biased since the techniques that the authors used actually sounds familiar to me. I'll take this into consideration.
> >
> > 2. Why was the parameter expansion necessary? Does it reduce the error?
> >
> > 3. This addresses my question. Thanks.

---

> > > ### Author Response · Authors · 2018-11-20
> > > **regarding the parameter expansion**
> > >
> > > 2. We expanded the parameter space for XGBoost to give it a larger computational budget. This larger budget compensates for the fact that fitting a single model using XGBoost is quicker than with Feat. The extra tuning made the XGBoost results slightly better; in the pdfdiff for Figure 3 of the revision, one can see a slight improvement in the boxplot for XGBoost. However, XGBoost's accuracy was still not significantly different than Feat over all problems (p = 1.0). Interestingly, the new XGBoost results did significantly outperform MLP, unlike the original results.

---

### Official Review · AnonReviewer1 · 2018-11-02
**A solid method for learning interpretable networks, though with a large computational cost**

**Rating:** 7
**Confidence:** 1

**Review:**

# Summary
The paper presents a method for learning network architectures for regression tasks. The focus is on learning interpretable representations of networks by enforcing a concise structure made from simple functions and logical operators. The method is evaluated on a very large number of regression tasks (99 problems) and is found to yield very competitive performance.

# Quality
The quality of the paper is high. The method is described in detail and differences to previous work are clearly stated. Competing methods have been evaluated in a fair way with reasonable hyperparameter tuning.

It is very good to see a focus on interpretability. The proposed method is computationally heavy, as can be seen from figure 7 in the appendix, but I see the interpretability as the main benefit of the method. Since many applications, for which interpretability is key, can bear the additional computational cost, I would not consider this a major drawback. However, it would be fair to mention this point in the main paper.

# Clarity
The paper reads well and is nicely structured. The figures and illustrations are easy to read and understand.

# Originality
The paper builds on a large corpus of previous research, but the novelties are clearly outlined in section 3. However, the presented method is very far from my own field of research, so I find it difficult to judge exactly how novel it is.

# Significance
The proposed method should be interesting to a wide cross-disciplinary audience and the paper is clearly solid work. The focus on interpretability fits well with the current trends in machine learning. However, the method is far from my area of expertise, so I find it difficult to judge the significance.

---

> ### Author Response · Authors · 2018-11-15
> **Computational tradeoff now discussed**
>
> We thank the reviewer for their positive comments. We agree with the reviewer's assessment of the tradeoff between interpretability and computational cost. Many applications with interpretability as a main focus can stand the additional burden (in this case, 60 minutes maximum). It is also worth noting that this method is parallelizable, although that functionality has not been exploited in our benchmarking.
>
> Based on the reviewer's comments and other comments, we have made the following changes:
>
> - we explicitly mention the termination criteria in the experiments section and the computation times in the results
>  - a discussion of the tradeoff of computational cost has been added to the discussion
>  - we have added a validation loss terminal criterion (a.k.a. early stopping) to Feat to improve the runtimes a bit
>
> Thanks for the helpful comments.

---

### Meta-Review · Area_Chair1 · 2018-12-16
**Well written paper on learning concise representations for regression with strong empirical evaluation**

**Confidence:** 4
**Recommendation:** Accept (Poster)

**Metareview:**

The reviewers all feel that the paper should be accepted to the conference.  The main strengths that they noted were the quality of writing, the wide applicability of the proposed method and the strength of the empirical evaluation.  It's nice to see experiments across a large number of problems (100), with corresponding code, where baselines were hyperparameter tuned as well.  This helps to give some assurance that the method will generalize to new problems and datasets.    Some weaknesses noted by the reviewers were computational cost (the method is significantly slower than the baselines) and they weren't entirely convinced that having more concise representations would directly lead to the claimed interpretability of the approach.  Nevertheless, they found it would make for a solid contribution to the conference.